# Viral respiratory infections and the oropharyngeal bacterial microbiota in acutely wheezing children

Leah Cuthbertson[1☯]*, Stephen W. C. Oo[2,3☯], Michael J. Cox[1], Siew-Kim Khoo[2,4], Des W. Cox[2], Glenys Chidlow[5], Kimberley Franks[2,4], Franciska Prastanti[2,4], Meredith L. Borland[2,6,7], James E. Gern[8], David W. Smith[2,5,9], Joelene A. Bizzintino[2,4], Ingrid A. Laing[2,4], Peter N. Le Souëf[2,4‡], Miriam F. Moffatt[1‡], William O. C. Cookson[1,10‡]

**1** National Heart and Lung Institute, Imperial College, London, England, United Kingdom, **2** Division of Paediatrics, Faculty of Health and Medical Sciences, University of Western Australia, Perth, Australia, **3** Respiratory Department, Perth Children's Hospital, Perth, Western Australia, **4** Telethon Kids Institute, Perth, Australia, **5** Department of Microbiology, PathWest Laboratory Medicine WA, QEII Medical Centre, Perth, Australia, **6** Emergency Department, Perth Children's Hospital, Perth, Australia, **7** Division of Emergency Medicine, Faculty of Health and Medical Sciences, University of Western Australia, Perth, Australia, **8** Department of Pediatrics, University of Wisconsin-Madison, Madison, Wisconsin, United States of America, **9** Medical School, Faculty of Health and Medical Sciences, University of Western Australia, Perth, Australia, **10** Royal Brompton and Harefield NHS Foundation Trust, London, England, United Kingdom

☯ These authors contributed equally to this work.
‡ These authors also contributed equally to this work.
* l.cuthbertson@imperial.ac.uk

**Data Availability Statement:** Sequences were submitted to the European nucleotide database, project number PRJEB32061.

## Abstract

Acute viral wheeze in children is a major cause of hospitalisation and a major risk factor for the development of asthma. However, the role of the respiratory tract microbiome in the development of acute wheeze is unclear. To investigate whether severe wheezing episodes in children are associated with bacterial dysbiosis in the respiratory tract, oropharyngeal swabs were collected from 109 children with acute wheezing attending the only tertiary paediatric hospital in Perth, Australia. The bacterial community from these samples was explored using next generation sequencing and compared to samples from 75 non-wheezing controls. No significant difference in bacterial diversity was observed between samples from those with wheeze and healthy controls. Within the wheezing group, attendance at kindergarten or preschool was however, associated with increased bacterial diversity. Rhinovirus (RV) infection did not have a significant effect on bacterial community composition. A significant difference in bacterial richness was observed between children with RV-A and RV-C infection, however this is likely due to the differences in age group between the patient cohorts. The bacterial community within the oropharynx was found to be diverse and heterogeneous. Age and attendance at day care or kindergarten were important factors in driving bacterial diversity. However, wheeze and viral infection were not found to significantly relate to the bacterial community. Bacterial airway microbiome is highly variable in early life and its role in wheeze remains less clear than viral influences.

**Funding:** This study was funded by an NHMRC program grant (#458513), NHMRC project grant (#1045760), the West Australian Institute of Medical Research and the Asthma Foundation of Western Australia (AFWA), the Wellcome Trust (P40069) and the Asmarley Trust. Fellowship support was also provided for SWCO (Telethon Research Fellowship and Princess Margaret Hospital Foundation Clinical Fellowship).

**Competing interests:** The authors have declared that no competing interests exist.

## Introduction

Respiratory infections are a significant cause of morbidity in young children, with over half of hospitalisations in children under 5 years of age being related to respiratory disease [1]. Evidence suggests that environmental and infectious exposures in early life are a major risk factor for the development of disease [2, 3], including asthma [4–6].

Acute wheeze is the most common reason for children to present to hospital with between 80–90% of these cases being attributed to viral infection [7, 8]. While a number of respiratory viruses have been implicated in wheeze the most common are respiratory syncytial virus (RSV) and rhinovirus (RV)[9]. In infants aged less than one year RSV infections predominate, but after this RV is also viewed as an important risk factor for acute wheeze [10].

The influence of the respiratory tract bacterial community in the context of wheeze is still being established. Recent studies in this area have implicated organisms such as *Moraxella catarrhalis*, *Haemophilus influenzae* and *Staphylococcus aureus* in increased rates of wheeze [11, 12]. It has been suggested this may be due to immune modulation by these organisms, in infants resulting in increased risk of asthma development in later life [13].

Using 16s rRNA gene sequencing of respiratory samples from children presenting to hospital with acute wheezing, this study aimed to examine whether the bacterial community in the airways of children with acute respiratory wheeze was altered compared with that of non-wheezing children. Changes in the bacterial community were also explored to determine if acute RV infection or species had a significant effect on the airway microbiota.

## Methods

Children between 0–16 years were recruited as part of the MAVRIC (Mechanisms of Acute Viral Respiratory Infection in Children) study on presentation to Princess Margaret Hospital (PMH) for Children, Perth, Western Australia between January 2004 and January 2014. Cases and controls were recruited through all seasons, and a questionnaire was administered to all subjects to determine symptoms of any current illness, including coryzal symptoms, and risk factors, including birth history, postnatal and in utero cigarette smoke exposure, daycare attendance, atopy and allergy history, diagnoses of acute illness (children were excluded if they had any chronic illness other than asthma), recent antibiotic use, systemic steroids.

Recurrence data was collected on cases included in the study from birth. Hospital presentation records were used to determine frequency of respiratory presentations both prior to and following presentation. Five patterns were determined described as "few", "persistent", "multiple A", "multiple B", and "atypical". (See supplemental methods for further definitions).

Cases recruited had acute wheezing illness with no other co-morbid conditions besides asthma, eczema, or atopy. Controls with no pre-existing chronic disease including chronic respiratory illness were recruited from four sources: siblings and relatives of cases, PMH patients (presenting with minor injury/fractured limbs), volunteers from the local community or day care facilities. A proportion of controls had symptoms of mild acute respiratory infection but no wheeze. Blood, oropharyngeal (OP) swabs and nasal samples (wash or blow) were collected from each participant.

Several cases were followed up within 9 months of recruitment and viral, OP swab and blood samples were repeated at this time.

This study was approved by the PMH for Children, Perth, Western Australia Human ethics committee (Reference: 1761EP). At least one parent or guardian provided informed written consent for children, prior to participation in the study.

## Atopy

Skin prick tests were performed to 11 allergens (cow's milk, whole egg, cat pelt, dog dander, rye grass, mixed grasses, *D. pteronyssinus*, *D. farinae*, cockroach and *Aspergillus fumigatus*, *Alternaria tenuis*), a wheal size ≥3mm or self-reported allergic reaction to an allergen was used determined atopy.

## Bloods cell counts and cathelicidin measurements

Blood cell counts were determined by PathWest Laboratory Medicine WA (PathWest) at PMH hospital pathology lab.

Cathelicidin (LL-37) was measured in plasma using a commercially available ELISA kit (Hycult Biotech).

## Viral detection

A nasal blow or wash was collected at recruitment and placed immediately on ice prior to storage at -80˚C. Nasal specimens were typed for RV species using methods previously described [14] using modified primers [15]. Briefly, a semi-nested RT-PCR with primers that amplify the 260-bp variable region of the 5' untranslated region of the RV genome was completed. RV-positive samples were sequenced and assigned an RV strain type and species following sequence alignment with sequences of the 101 classic serotypes and the 53 newly assigned genotypes, using ClustalX software (University College Dublin, Dublin, Ireland) [16].

A second aliquot from each nasal specimen was tested for RSV, influenza A and B, parainfluenza 1–4, RSV, and human metapneumovirus (hMPV) at PathWest using routine diagnostic methods including PCR [17], direct or indirect fluorescent antibody testing, or immunofluorescence after cell culture as previously described [18]. Whenever possible, samples were also tested for enterovirus, coronavirus, and bocavirus.

## Bacterial detection

Sterile dry rayon swabs (Copan) were used to sample the soft palate of the oropharynx. Specific care was made not to contaminate samples with any other part of the mouth. Any swab that made contact with the tongue or cheek were disposed of and a new swab was performed, for full details see supplementary materials. Samples were immediately put on ice and stored at -80˚C.

DNA was extracted from samples utilizing an MPbio FastDNA SPIN Kit for soil as per manufacturers' instructions and frozen at -80˚C. Samples were transported on dry ice to Imperial College London, for bacterial analysis.

## Quantitative PCR

Total bacterial burden was measured using a SYBR green quantitative PCR assay using the primers 520F, 5'– `AYTGGGYDTAAAGNG` and 820R, 5'–`TACNVGGGTATCTAATCC`, targeting the V4 region of the 16S rRNA gene as described in Cuthbertson *et al* 2017[19]. All reactions were performed in triplicate and included standards and non-template controls on the ViiA 7 Real-time PCR system (Life Technologies, Paisley, UK) using SYBR Fast qPCR Master mix (KAPA Biosystems, Wilmington, MA, USA). Standards were generated from near full length cloned 16S rRNA gene of *Vibrio natregens*. Plasmids quantified using Quantit picogreen dsDNA Assay kit (Promega, Madison, USA), samples were then serially diluted 10 fold to form standards ranging from $1 \times 10^8$–$1 \times 10^4$.

## 16S rRNA sequencing

Community analysis was carried out using 16S rRNA gene sequencing. Custom dual barcoded fusion primers were used to target the previously quantified region of the 16S rRNA gene as previously described [19]. Each sequencing run contained a PCR negative control and a Mock community, consisting of 34 16S rRNA gene clones of known bacterial species in equal proportions. Sequencing was carried out using the Illumina MiSeq platform using the Illumina V2 2x250bp cycle kit. Sequences were submitted to the European nucleotide database, project number PRJEB32061.

## Sequencing analysis

Downstream sequencing analysis was carried out using Quantitative Insights in Microbial Ecology (QIIME) Version 1.9.0 due to the dual barcoded indexes used in this study. All sequences were trimmed to 200bp and joined with a minimum of 150bp overlap, a maximum of 10% mismatch was stipulated. Sequences were then demultiplexed and any phiX reads were removed, prior to OTU picking using open reference UCLUST OTU picking [20], clustering at 97% similarity using the Silva reference database (www.arb-silva.de), reads with less that 60% id were discarded and 10% of sequences that failed to id were included for de novo clustering. Representative sequences were picked from the most abundant read in the cluster. PYNAST [21] was used to align representative sequences before running the nearest alignment space termination (NAST) algorithm [22]. ChimeraSlayer (http://microbiomeutil.sourceforge.net/) was used to identify and remove any chimeric sequences. The Ribosomal Database Project (RDP) naive Bayesian classifier was used to apply taxonomic identification using Silva 115 NR database. Finally, an OTU biom table was created for further downstream analysis.

Exact sequence variants (ESVs) may in some circumstances improve identification of microbial taxa, but genomic sequencing of the airway microbiota is at an early stage. In order to avoid over-splitting of taxa and false inflation of diversity, we have taken the conservative approach of using OTUs in our analyses rather than amplicon sequence variants.

## Statistical analysis

All further analysis was carried out using R version 3.3.2 [23]. Pre-processing and primary analysis was carried out in Phyloseq [24]. Contamination was removed using Decontam [25]. A minimum threshold of 2,000 reads was applied to all, samples with less than 2,000 reads were removed from further analysis. All remaining samples were then rarefied to the minimum number of reads present in the data subset.

Non-parametric Wilcoxon sign ranked tests were used to test significant differences between means. Pearson correlations were used to test the relationship between continuous clinical variables and diversity measures. Adonis permutational ANOVA was used to investigate changes in community composition while Random forest analysis was used to identify OTU's associated with disease.

## Results

After removal of sequencing controls and those with less than 2000 reads, a total of 201 samples were taken forward for analysis, see Table 1 (further information in S1 Table). This included 109 samples from children with acute wheeze (Fig 1) and paired stable follow-up samples from 17 of these children. Control samples from 75 children without symptoms of wheeze were collected, this cohort included children from day-care and siblings of those with acute wheeze (Fig 1).

**Table 1. Demographic table of acute cases and healthy controls.** Data indicates counts for cases (acute wheeze diagnosis) and healthy controls. N indicates the total number of subjects that information was obtained for each group. Count data is indicted as a count (% positive of N). Continuous data is recorded at median of N (mix–max). Differences in clinical variables were calculated using Wilcoxon sign rank test, differences in ethnic groups, sampling season an RV type were calculated using Chi-squared.

| | | Acute | | Controls | Statistical comparison |
|---|---|---|---|---|---|
| | **N** | **All** | **N** | **All** | **p** |
| Female (male) | 109 | 51 (58) | 75 | 38 (36) | 0.546 |
| Age (median (Min-Max)) | 109 | 3.83 (0.08–13.93) | 75 | 3.155 (0.8–18.5) | 0.344 |
| Gestation period (Min-Max) | 109 | 40 (38–43) | 65 | 39 (35–42.5) | 0.095 |
| **Ethnic group** | | | | | **0.057** |
| Aboriginal (%) | 109 | 5 (5%) | 70 | 2 (3%) | |
| African/African American (%) | | 12 (11%) | | 1 (1%) | |
| Asian/Indian (%) | | 23 (21%) | | 16 (21%) | |
| Caucasian (%) | | 54 (50%) | | 51 (68%) | |
| Maori (%) | | 3 (3%) | | 0 | |
| Pacific Islanders/Samoan (%) | | 2 (2%) | | 0 | |
| PNG (%) | | 2 (2%) | | 0 | |
| Undetermined (%) | | 8 (7%) | | 5 (7%) | |
| **Season** | | | | | **0.803** |
| Autum (%) | 109 | 20 (18%) | 75 | 4 (5%) | |
| Spring (%) | | 18 (16%) | | 17 (23%) | |
| Summer (%) | | 3 (3%) | | 3 (4%) | |
| Winter (%) | | 68 (62%) | | 51 (68%) | |
| **Diagnosis** | xs | | | | |
| Asthma exacerbation (%) | 109 | 37 (34%) | 27 | 0 | <0.001 |
| Viral wheeze (%) | 109 | 49 (45%) | 27 | 0 | <0.001 |
| Bronchiolitis (%) | 109 | 14 (13%) | 27 | 0 | 0.05 |
| Pneumonia (%) | 109 | 1 (0.9%) | 27 | 0 | 0.632 |
| URTI (%) | 109 | 94 (86%) | 64 | 29 (45%) | <0.001 |
| Atopy (%) | 89 | 54 (61%) | 63 | 21 (33%) | <0.001 |
| Severity Zscore (min-max) | 85 | 0.3191 (-2.2306–2.0151) | | | NA |
| **Medication** | | | | | |
| Systemic steroids (%) | 105 | 76 (72%) | 67 | 0 | <0.001 |
| Oxygen (%) | 99 | 45 (45%) | | | NA |
| **Blood counts** | | | | | |
| Platelets (min-max) | 82 | 295 (105–611) | 41 | 292 (35–655) | 0.996 |
| T-cells (min-max) | 83 | 10.5 (3.5–27.9) | 41 | 8.3 (4–13.4) | 0.006 |
| Neutrophils (min-max) | 83 | 6.87 (0.47–20.41) | 40 | 3.325 (0.71–8.19) | <0.001 |
| Lymphocytes (min-max) | 83 | 1.67 (0.27–8.61) | 40 | 3.74 (1.17–7.94) | <0.001 |
| Monocytes (min-max) | 83 | 0.54 (0.275–3.39) | 40 | 0.685 (0.3–2.18) | 0.019 |
| Eosinophils (min-max) | 83 | 0.06 (0.015–2.67) | 40 | 0.28 (0–1.42) | <0.001 |
| Basophils (min-max) | 83 | 0.01 (0–0.38) | 40 | 0.05 (0–0.18) | <0.001 |
| tven1 | 81 | 2 (0.8–100.4) | | | NA |
| O2 Saturation | 89 | 95 (69–99) | | | NA |
| ll37 | 60 | 2.0986 (0.2404–6.7606) | | | NA |
| **Viral infection** | | | | | |
| RV | 109 | 71 (65%) | 75 | 33 (43.99%) | 0.004 |
| RV group | 71 | | | | **0.009** |
| A | | 26 (37%) | | 17 (51.51%) | |
| B | | 3 (4%) | | 6 (18.18%) | |

*(Continued)*

**Table 1.** (Continued)

| | | Acute | | Controls | Statistical comparison |
|---|---|---|---|---|---|
| | N | All | N | All | p |
| C | | 40 (56%) | | 10 (30.3%) | |
| Unknown | | 2 (3%) | | | |
| RSV | 76 | 16 (215) | 63 | 2 (3%) | 0.002 |
| Adenovirus | 76 | 3 (4%) | 63 | 6 (10%) | 0.187 |
| Influenza virus | 76 | 0 (0%) | 63 | 1 (2%) | 0.278 |
| Parainfluenza virus | 76 | 1 (1%) | 63 | 2 (3%) | 0.459 |
| Corona virus | 39 | 20 (51%) | 63 | 1 (2%) | 0.312 |
| Human metapneumovirus | 75 | 4 (5%) | 63 | 0 | 0.065 |
| Enterovirus | 34 | 3 (9%) | 9 | 3 (33%) | 0.066 |
| Bocavirus | 5 | 2 (40%) | 51 | 6 (12%) | 0.092 |
| **Smoking** | | | | | |
| Mother smoked ever | 108 | 38 (35%) | 75 | 15 (20%) | 0.023 |
| Smoking now | 108 | 18 (17%) | 75 | 2 (3%) | 0.003 |
| Smoked when pregnant | 106 | 18 (16%) | 75 | 3 (4%) | 0.007 |
| smoked regularly during pregnancy | 106 | 15(14%) | 75 | 1 (1%) | 0.003 |
| Anyone smoke in the house? | 109 | 33 (30%) | 75 | 10 (13%) | 0.008 |
| **Siblings and school** | | | | | |
| No of Children | 107 | 2 (0–5) | 75 | 2 (0–5) | 0.028 |
| No of siblings | 108 | 1 (0–8) | 72 | 0 (0–3) | 0.015 |
| Kindergarten | 107 | 52 (49%) | 74 | 18 (24%) | 0.001 |
| Pre-school | 107 | 42 (39%) | 74 | 9 (12%) | <0.001 |

## Comparison of acute wheeze and controls

Samples from 109 children with acute wheeze and 75 control children were compared (Table 1 and Fig 1). No significant difference in alpha diversity was observed between the 2 groups (Richness, P = 0.363; Shannon-Weiner, P = 0.98; Inverse Simpsons, P = 0.654). Adonis permutational ANOVA revealed a significant difference in Bray-Curtis dissimilarity ($R^2$ = 0.016, P = 0.003), Fig 2A. However, the variation explained was only 1%, suggesting a relationship driven by a small number of samples (see S1 Fig).

A random forest model was used to predict wheeze based on OTUs in the bacterial community. The predictive power of the model for cases and controls was found to be poor (out of box (OOB) error rate = 34.24%, p = 0.37), therefore the model results were considered unreliable and was not used in further investigation. However, indicator species analysis revealed a significant association of a single *Veillonella* OTU with the acute group. DeSeq2 analysis indicated the same *Veillonella* OTU associated with cases while several *Haemophilus* OTUs were found to be associated with controls (S2 Fig)

Clinical variables were investigated to see if they were able to better explain bacterial community variation within this population. Wilcoxon rank sum test was used to investigate 33 binary variables across all samples (S2 Table). After Bonferroni correction for multiple testing, only the diagnosis of bronchiolitis was found to show a significant difference in bacterial richness (p = 0.006), Shannon-Weiner (p = 0.017) and inverse Simpsons (p = 0.062). Significant differences in Shannon-Weiner and inverse Simpsons were observed in those that required oxygen and those that had ever attended kindergarten. It was noted, however, that none of the controls were positive for bronchiolitis or required oxygen. Therefore, this relationship was further tested in the acute wheeze population. Kruskal-Wallis with Dunns test was used to

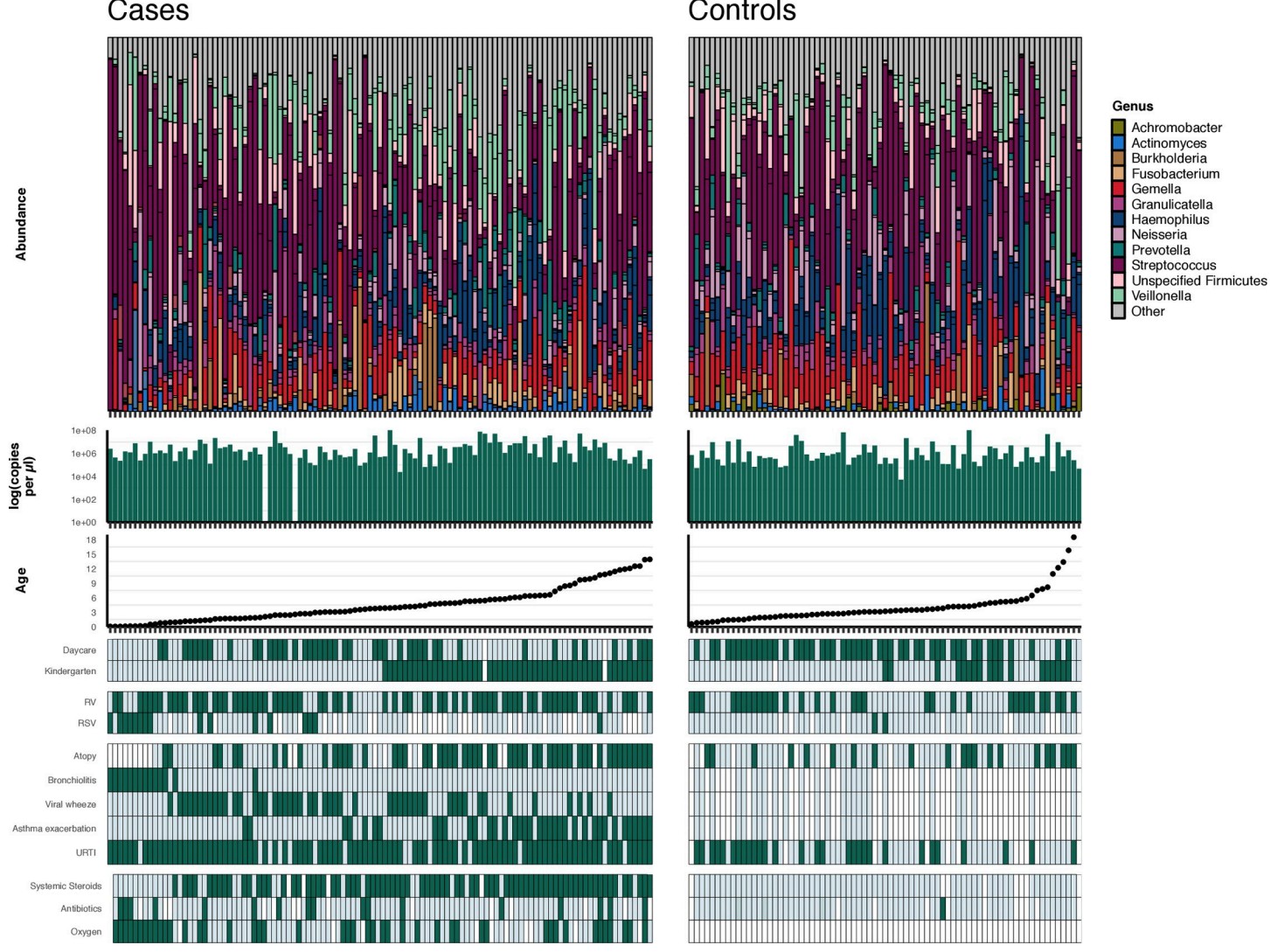

**Fig 1. Stacked bar plots of cases (acute wheeze) and controls ordered by age.** Bacterial biomass and patient demographics are indicated in the plots below. Dark green indicates yes while pale green indicates no, white boxes indicate NA or missing information.

investigate categorical variables, however no significant results were observed. We did not find any differences in the microbiota that were attributable to recorded breastfeeding. No data was available to explore the effect of delivery method in this study.

Pearson's correlations were used to investigate relationships between 20 continuous variables and alpha diversity measures (S3 Table). Only bacterial biomass was found to have a weak significant correlation with bacterial richness. Significant changes in blood counts were not found to hold up to multiple testing and expected differences in the antimicrobial peptide cathlecidin were not found.

Adonis permutational ANOVA with 99,999 iterations was used to investigate changes in community composition with 58 variables. Only bacterial biomass, bronchiolitis diagnosis and regular attendance at day care were found to be significant (S4 Table).

Bronchiolitis diagnosis was found to show significant reduction in alpha diversity. Due to the control samples coming from a slightly older population true age matched case controls

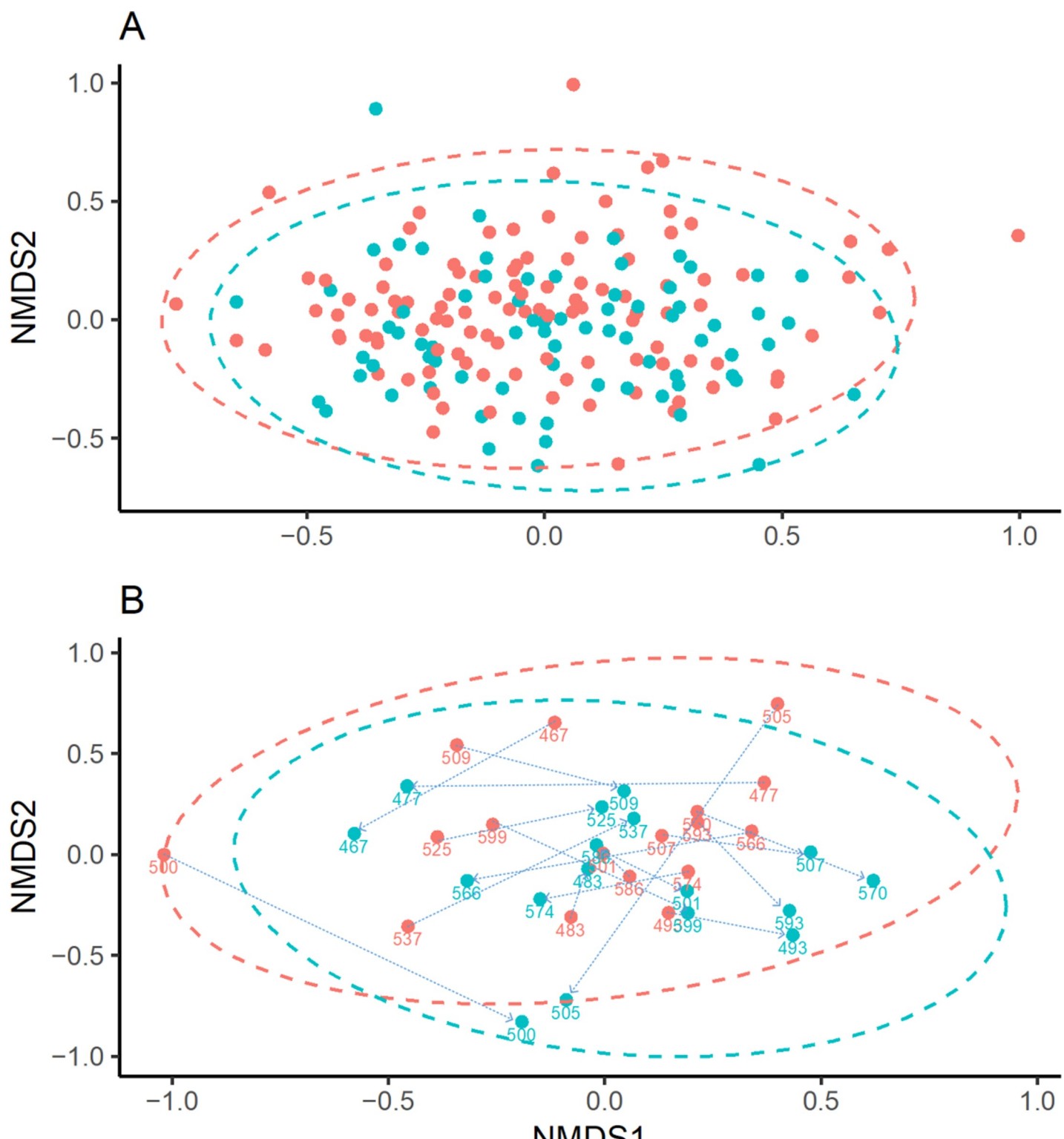

**Fig 2. NMDS plots based on Bray-Curtis dissimilarity comparing.** A) non-wheezing controls (blue) to acute wheezing children (red), $R^2 = 0.016$. B) Wheezing children (red) and paired follow-up samples (blue). Numbers indicate paired samples from individual patients and are linked by arrows. Ellipses were added assuming a multivariate normal distribution.

were unable to be compared (wheeze, median = 0.195 (min = 0.08 –max = 1.87), control, 1.38 (0.6–1.91)). However, when the microbiota in children with bronchiolitis were compared to those closest in age there was a significant reduction in alpha diversity (richness; p = 0.01, Shannon-Weiner, p = 0.003, inverse Simpsons, p = 0.005) that was not associated with change in bacterial biomass (p = 0.874). Changes in community composition using Bray-Curtis dissimilarity were found to explain 9.8% of the variation (p = 0.012), S3 Fig.

## Acute wheeze

To explore variation within the group suffering from acute wheeze, this group was examined as a subset from the whole data set and clinical variables were explored (Fig 1). In the acute wheeze group, significant decrease in all alpha diversity measures (differences in bacterial diversity between subjects) were seen with bronchiolitis. Fourteen patients with doctor diagnosed bronchiolitis were included in the study, all of these patients were under the age of 2 years. Of those diagnosed, all patients also had wheeze. Nine of the patients were positive for RSV at the time of sampling, 5 of these were also positive for RV (Fig 1).

A significant increase was observed with attendance at kindergarten (S5 Table). Attendance at preschool showed a significant increase in bacterial richness (number of OTUs) only. Only patient age was found to show significant positive correlation with bacterial richness (S6 Table).

In the children with acute wheeze, bacterial biomass was found to explain 8.9% of the variation in beta diversity (p < 0.001). Bronchiolitis ($r^2$ = 0.042, p < 0.001) and attendance at kindergarten ($r^2$ = 0.052, p < 0.001) were also found to be significant in the acute wheeze dataset (S7 Table).

Within the 109 children with acute wheeze, 71 were below the age of 5. In the acute wheeze group, the under 5s bacterial biomass remained a significant driver of bacterial community composition ($r^2$ = 0.06, p < 0.001). However, no other variables were significant in the younger cohort.

Kruskal-Wallis was used to investigate wheeze recurrence patterns for the five patterns of wheeze observed in wheeze cases (S4 Fig). No significant difference in alpha bacterial diversity was observed within or between any of the above recurrence groups.

## Rhinovirus (RV)

Within the participants suffering from acute wheeze, 71 patients had RV infection. RV strains from two patients were unable to be typed. Twenty-six patients were infected with RV-A, 3 with RV-B and 40 with RV-C. The sample numbers only allowed comparison of the bacterial community from patients with RV-A with those with RV-C. No significant difference in alpha diversity was observed. In the under 5s group, 15 patients had RV-A, 1 had RV-B and 27 had RV-C. A significant difference in richness (W = 123.5, p-value = 0.039) was observed between those with RV-A and RV-C, however no differences in diversity were observed. No significant difference in community composition was observed using adonis ($R^2$ = 0.039, p = 0.328).

Random forest modelling was unable to identify OTUs associated with viral species (p = 0.504).

## Follow-up samples

Stable follow-up samples were collected from 17 individuals after an acute wheezing episode. No significant difference in bacterial biomass (V = 44, p-value = 0.132) or alpha diversity (richness, p = 0.289, Shannon, p = 0.145, Simpson, p = 0.109) was found when comparing paired samples with Wilcoxon signed rank test.

Between and within sample beta diversity was compared (Fig 2B). Significant differences between paired samples were observed with 5.8% of the variation explained by time of sampling (p = 0.013), however 57.6% of the variation was explained by individual (p = 0.013).

## Discussion

In this cohort of children, we found substantial diversity and heterogeneity in OP microbiota composition regardless of wheeze status. There were no significant microbiome differences between acute wheeze cases and non-wheezing controls on alpha diversity measures. Beta diversity was examined using a Bray-Curtis dissimilarity adonis model and while significant difference (p = 0.003) was found, only 1.6% of the variance is explained by differences between cases and controls. Further interrogation with Bray-Curtis dissimilarity hierarchical clustering (S1 Fig) reveals that the correlation is likely driven by a small number of samples and no pattern was observed between those that had wheeze and those that did not (Fig 2). Overall, despite cases having an acute wheezing illness serious enough to cause presentation to a children's hospital emergency department, there were no clear differences in the bacterial community between cases and controls.

In the small number of cases with matched acute and convalescent samples taken up to 9 months later, we were similarly unable to find differences in alpha diversity between the cases and follow-up samples. Significant differences in beta diversity were observed between cases and follow-up samples, however the variation explained was 5.8% compared to within subject changes which explained 57.6% of the variation. The heterogeneous nature of the microbial community between patients makes investigation into microbial changes a challenge and supports the need for large-scale longitudinal investigation into community composition in health and disease.

Random forest analysis has been used in a number of recent studies to investigate predictive OTUs in microbiome analysis [26, 27]. In this study, random forest was used to attempt to identify OTU predictors for wheeze. We found that the variation in the microbial community observed between these groups was high and this led to low confidence in the predictive power of the model. Random forest analysis of the bacterial community found bacterial OTUs to be poor predictors of wheeze. This analysis was also performed to determine if any OTUs were significantly associated with RV infection or between RV species. Neither model was found to have strong predictive power and estimated accuracy was low.

Diagnosis with bronchiolitis showed a significant difference in bacterial richness compared with the entire cohort, however this comprised only a small subset of patients (n = 13). When those with a diagnosis of bronchiolitis were compared to other cases that wheezed or age matched control subjects, significant differences in alpha and beta diversity were observed however this may have been driven by subjects with bronchiolitis being significantly younger than other groups of subjects. Bacterial diversity was found to be significantly reduced in these subjects with many patients dominated with *Streptococcus sp*. Further investigation into the streptococcal species, which is unable to be elucidated from 16S rRNA sequencing, would be important in further investigations. Longitudinal studies in young children to determine if low bacterial diversity in the respiratory tract is a risk factor for or the result of bronchiolitis would be important for future investigations.

In contrast to our findings, there are a number of studies that show significant differences in microbiome in relation to viral infections compared to healthy subjects [11, 28–30]. These studies show haemophilus, streptococcus, moraxella associated with acute respiratory tract infections and staphylococcus, alloiococcus and corynebacterium in healthy samples. A number of factors account for the differences we find; firstly, they analyse nasopharyngeal (NP)

samples compared to the OP samples analysed in our study. A number of studies have shown that these areas show significantly different bacterial communities, and that OP swabs are a better proxy for the lower respiratory tract microbiome. There are also age differences between other cohorts and ours.

Lastly analytical methods to look at the microbiome differ between studies. Analytical methods for microbiota are slowly being standardised, and much of the analytic techniques are borrowed from ecological analysis to try and accurately portray biodiversity and differences. In order to find differences, earlier studies have defaulted to simplistic groupings based on dominant organisms. However, we have observed that many samples are diverse and without a singular dominant organism. For example, Man *et al*, comparing those with lower respiratory tract infection and healthy controls, found no difference in alpha diversity, and using beta diversity measures found, similar to us, very small but significant differences between the groups ($r^2$ = 0.0031, p = <0.0001) [28]. To find significant differences in species, previous studies grouped samples based on perceived dominance and not necessarily by a dominance measure or beta diversity based on hierarchical clustering.

The greater variation in age within our population compared to other studies may have influenced our ability to find significant differences. Bacterial biomass was found to explain much of the variation in beta diversity within the acute population. Bacterial biomass was also found to show a weak positive correlation with age. Age was found to be a key variable in the analysis of this dataset and this relationship has been demonstrated in previous studies [31]. Age is interrelated with developmental changes in blood cell counts and immune function [32]. With these considerations, we considered changes in the microbiome while controlling for age.

A strength of our study is that we extracted blood from patients to examine differences in blood counts and antimicrobial peptides. However, variation in age in the study resulted in small and non-significant trends in blood cell counts. Cathelicidin was also examined as a candidate innate immunity marker as it has both anti-viral and anti-bacterial activity and lower levels have been shown to be associated with worsened severity in bronchiolitis [33]. However, we were unable to find a significant change in the bacterial community with changes in cathelicidin.

Seasonal influences in the same study location (Perth, Australia) have shown to have limited effects on microbiome in comprehensive longitudinal studies at least in infants [29]. Risk factors that are associated with hospitalisation as a result of respiratory infections in early life include maternal smoking during pregnancy, season of birth, delivery mode and gestational age[34]. When considered in this cohort of mostly hospitalised wheezing children (96%), these factors were not found to be significantly associated with bacterial or viral infection suggesting more work is required to fully understand any potential relationship.

At least one other study exists comparing OP microbiome samples between healthy adult patients and severe asthmatics showing no significant diversity differences [35]. An abundance of moraxella is found in prior studies in NP. Hilty *et al* demonstrate that Moraxella (proteobacteria) exists in oropharynx and may account for differences [36]. OP samples were specifically chosen as they showed greater representation to lower airway samples at least in stable patients [36, 37]. Whether acute OP swabs reflect lower airway samples in acute wheeze or acute viral episodes is still unclear but, interestingly, Man *et al* did show that intensive care patients had reasonable similarity between NP samples and lower respiratory tracheal samples (Bray Curtis similarity p = 0.61), but did find key differences, specifically that staphylococcus, Corynebacterium, and Dolosigranulum sp. were almost exclusively present in NP samples and absent from endotracheal aspirates [28].

The bacterial community in the paediatric population is diverse and heterogeneous. The wide range of clinical factors tested did not fully explain the wide variation in the bacterial community in these subjects. Age had a significant influence on both the bacterial community and blood cell counts in this study and the wide age range within the study population helped explain some of the variation observed.

The simplest interpretation of our results is that acute wheezing illnesses are driven by viral infections and that these infections have little influence on the bacterial community during the acute phase of the illness. However, the heterogeneous nature of the subjects made it difficult to test for significant associations between clinical variables and the oropharyngeal bacterial community. Prospective longitudinal investigation of children pre, during and post viral infection may help identify if the bacterial community is either protective or a risk factor for viral infection and respiratory wheeze.

## Supporting information

**S1 Table. Participant demographics continued.** Data is recorded as n (%) or median (min-max). Data is recorded as a percentage of all data collected.
(DOCX)

**S2 Table. Comparison of categorical clinical variables with Wilcoxon rank sum test in the complete cohort to alpha diversity measures, richness, Shannon-Weiner and inverse Simpsons.** P values adjusted using Bonferonni correction for multiple testing.
(DOCX)

**S3 Table. Pearson's correlations of continuous clinical variables and alpha diversity measures, richness, Shannon-Weiner and inverse Simpsons, in the complete cohort.** P values adjusted using Bonferonni correction for multiple testing.
(DOCX)

**S4 Table. Results of Bray-Curtis adonis permutational ANOVA examining clinical variables with 99,999 iterations.** P values adjusted using Bonferonni correction for multiple testing.
(DOCX)

**S5 Table. Comparison of categorical clinical variables in those with acute wheeze to alpha diversity measures, richness, Shannon-Weiner and inverse Simpsons.** P values adjusted using Bonferonni correction for multiple testing.
(DOCX)

**S6 Table. Pearson's correlations of continuous clinical variables in those with acute wheeze and alpha diversity measures.** richness, Shannon-Weiner and inverse Simpsons. P values adjusted using Bonferonni correction for multiple testing.
(DOCX)

**S7 Table. Results of Bray-Curtis adonis permutational ANOVA examining clinical variables from individuals with acute wheeze with 99,999 iterations.** P values adjusted using Bonferonni correction for multiple testing.
(DOCX)

**S1 Fig. Stacked bar plot comparing OP samples from children with acute wheeze to healthy controls.** Hierarchical clustering based on Bray-Curtis dissimilarity was used to order stacked bar plots for individuals. Adonis permutational ANOVA explained 1.6% of the variation.
(TIFF)

**S2 Fig. Volcano plot showing differential abundance of OTUs based on results from DeSeq2 analysis.** OTUs considered statistically significant (p-value < 0.001) were coloured based on genus level identification.
(PDF)

**S3 Fig. Stacked bar plot comparing OP samples from children with bronchiolitis and acute wheeze to healthy controls.** Hierarchical clustering based on Bray-Curtis dissimilarity was used to order stacked bar plots for individuals. Adonis permutational ANOVA revealed bronchiolitis explained 9.8% of the variation.
(TIFF)

**S4 Fig. Boxplots showing diversity measures associated with recurrence data from acute wheeze cases.**
(TIFF)

**S1 File. Supplementary methods.**
(DOCX)

## Author Contributions

**Conceptualization:** Joelene A. Bizzintino, Ingrid A. Laing, Peter N. Le Souëf, Miriam F. Moffatt, William O. C. Cookson.

**Data curation:** Stephen W. C. Oo, Siew-Kim Khoo, Des W. Cox, Kimberley Franks, Franciska Prastanti, Meredith L. Borland, Ingrid A. Laing.

**Formal analysis:** Leah Cuthbertson.

**Investigation:** Stephen W. C. Oo, Des W. Cox, Glenys Chidlow, Kimberley Franks, James E. Gern, David W. Smith.

**Methodology:** Leah Cuthbertson, Stephen W. C. Oo, Michael J. Cox, James E. Gern, David W. Smith.

**Project administration:** Ingrid A. Laing.

**Supervision:** Peter N. Le Souëf, Miriam F. Moffatt, William O. C. Cookson.

**Writing – original draft:** Leah Cuthbertson.

**Writing – review & editing:** Leah Cuthbertson, Stephen W. C. Oo, Michael J. Cox, Peter N. Le Souëf, William O. C. Cookson.

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
