## [Decision Letter · Decision Letter 0]

30 Aug 2019

[EXSCINDED]

PONE-D-19-20152

Viral respiratory infections and the oropharyngeal bacterial microbiota in acutely wheezing children

PLOS ONE

Dear Dr Cuthbertson,

Your paper was reviewed by two experts in the field and their comments follow. You need to optimize the analysis and clarify whether the subjects are infected by RSV or another virus as suggested by the two reviewers. We invite you to submit a revised version of the manuscript that addresses these points raised during the review process.

We would appreciate receiving your revised manuscript by Oct 14 2019 11:59PM. To enhance the reproducibility of your results, we recommend that if applicable you deposit your laboratory protocols in protocols.io, where a protocol can be assigned its own identifier (DOI) such that it can be cited independently in the future. For instructions see: http://journals.plos.org/plosone/s/submission-guidelines#loc-laboratory-protocols

We look forward to receiving your revised manuscript.

Kind regards,

Dong-Yan Jin

Academic Editor

PLOS ONE

Journal Requirements:

Additional Editor Comments:

Revised paper will be re-reviewed by the original reviewers if they are available.

Reviewers' comments:

Reviewer's Responses to Questions

**Comments to the Author**

1. Is the manuscript technically sound, and do the data support the conclusions?

Reviewer #1: Partly

Reviewer #2: Yes

2. Has the statistical analysis been performed appropriately and rigorously? 

Reviewer #1: No

Reviewer #2: Yes

3. Have the authors made all data underlying the findings in their manuscript fully available?

Reviewer #1: Yes

Reviewer #2: Yes

4. Is the manuscript presented in an intelligible fashion and written in standard English?

Reviewer #1: Yes

Reviewer #2: Yes

5. Review Comments to the Author

Reviewer #1: Re. Viral respiratory infections and the oropharyngeal bacterial microbiota in acutely wheezing children

It is an interesting research study to investigate the association between oral microbiota and acute wheeze in young children, in which the role of oral microbiota in influencing the viral infection, or acute wheeze in changing oral microbial community remains to be estimated. 109 case samples from children with acute wheeze (also paired follow-ups from 17 of these children) and 75 control samples from children without symptoms of wheeze were collected and compared for oral microbiota community and viral infection. Relevant clinical information, such as gender, age, atopy, blood cell counts, and cathelicidin measurements, were analyzed in combination of oral bacterial and viral data.

However, this work could be significantly improved with optimized bioinformatics and biostatistical analyses. The following comments may have the authors attentions:

1. QIIME 1 has been succeeded by QIIME 2 because of outdated workflow of QC control and OTU clustering in QIIME 1. The authors should update their analyses by using QIIME 2 or other pipelines that is more advanced than QIIME 1.9.0. This would ideally use amplicon sequence variants instead of 97% OTUs (see this paper for a discussion of this topic: https://www.ncbi.nlm.nih.gov/pubmed/28731476). Meantime, key parameters of workflow for OTU clustering and taxa assignment would be clearly indicated.

2. The authors used 16S V4 region primers and quantitative PCR to measure oral bacterial biomass. If so, was there baseline control for normalization?

3. The authors used three key measures, including Richness, Shannon-Weiner and Inverse Simpsons, to compare the differences of oral microbiota communities between samples from children with and without acute wheeze. However, more detailed comparison of specific bacteria taxa should be performed to show the differential bacterial composition and abundance between cases and controls. For example, as shown in Figure 1, which bacteria(s) was/were the mostly changed in relative abundance; it could be estimated with LEfSe, Wilcoxon signed rank test or Turkey HSD etc.

4. The age should be one of important factors influencing oral microbial community. Given the wide range of years of children age (0.08-18.5 yeas), the authors should consider to divide the samples into different ago groups and compare the difference of bacterial composition and abundance.

5. Table 1 lacks statistical comparison of demographic information between cases and controls.

6. Figure 1 lacks the basic unit of Biomass. The percentage proportion of abundance should be indicated in the y-axis.

7. Figure 2 - is it possible that the authors try other PCoA method to cluster samples, for example, weighted or unweighted UniFrac methods?

8. The authors mentioned that "Specific care was made not to contaminate samples with any other part of the mouth". Was there more detailed information?

Reviewer #2: The article is very relevant and evaluated viral respiratory infections and the oropharyngeal bacterial microbiota in acutely

wheezing children. Although they concluded that the microbiota is not associated with viral infection and wheeze in children the article discuss important points that deserve to be published.

However, it not clear if the children with bronchiolitis are infected with RSV or other virus. This must be explored in the results and in the conclusion. Other important point is to performed and analysis of family and order of the microbiota comparing children with bronchiolitis and the controls.

Another point that should be explored in the manuscript is breastfeeding and cesarean delivery, that might be influencing the microbiota.

6. PLOS authors have the option to publish the peer review history of their article (what does this mean?). If published, this will include your full peer review and any attached files.

Reviewer #1: No

Reviewer #2: No

---

## [Author Response · Author response to Decision Letter 0]

17 Sep 2019

Dear Dr Jin, 

Thank you for your response to our manuscript. We greatly appreciate the reviewers taking the time to consider this paper for publication in PlosOne, and their very helpful suggestions. We have taken all of their remarks into consideration as follows.

Reviewer #1

We note that the Reviewer considers this to be an interesting research study that would be improved by additions to the bioinformatic and biostatistical analyses. Our responses to specific points are: 

1. QIIME 1 has been succeeded by QIIME 2 because of outdated workflow of QC control and OTU clustering in QIIME 1. The authors should update their analyses by using QIIME 2 or other pipelines that is more advanced than QIIME 1.9.0. This would ideally use amplicon sequence variants instead of 97% OTUs (see this paper for a discussion of this topic: https://www.ncbi.nlm.nih.gov/pubmed/28731476). Meantime, key parameters of workflow for OTU clustering and taxa assignment would be clearly indicated.

We are aware of the update of QIIME 1.9.0 to QIIME 2, however QIIME 2 does yet not support dual barcoded data. A work around for this is to use internal illumina demultiplexing, but this process involves the removal of reads pre-analysis and is not optimal. We therefore clarified this reason on line 163 in the methods. 

The Reviewer is quite right to draw attention to the potential use of amplicon sequence variants (ASVs). However, ASVs have not been universally accepted, and despite arguments for the use of exact sequence variants (ESVs), OTUs remain an appropriate method of analysis for 16S data. The most important reasons for using OTUs over ASVs or ESVs is that they may lead to the over splitting of OTUs and false inflation of diversity. Considering intragenomic heterogeneity it is possible that multiple ESVs may not be coming from distinct taxa. 

We therefore acknowledge this debate in line 177 in the discussion by stating “Exact sequence variants (ESVs) may in some circumstances improve identification of microbial taxa, but genomic sequencing of the airway microbiota is at an early stage. In order to avoid over-splitting of taxa and false inflation of diversity, we have taken the conservative approach of using OTUs in our analyses rather than amplicon sequence variants”. 

2. The authors used 16S V4 region primers and quantitative PCR to measure oral bacterial biomass. If so, was there baseline control for normalization?

The protocols for performing quantitative PCR (2) have been updated with edits to both the methods and supplementary methods. These updates clarify the procedure used performing oropharyngeal swabs and explain the internal controls implemented during qPCR analysis. Full details of the parameters used for sequencing analysis have been updated in the methods section of the paper on page7, line 148, referencing all parameters used in the analysis.

3. The authors used three key measures, including Richness, Shannon-Weiner and Inverse Simpsons, to compare the differences of oral microbiota communities between samples from children with and without acute wheeze. However, more detailed comparison of specific bacteria taxa should be performed to show the differential bacterial composition and abundance between cases and controls. For example, as shown in Figure 1, which bacteria(s) was/were the mostly changed in relative abundance; it could be estimated with LEfSe, Wilcoxon signed rank test or Turkey HSD etc.

We reported the use of Random Forest modelling in the paper to predict OTUs significantly associated with disease states. These models had low predictive power and the model results were not considered reliable (i.e. there is no great difference between the sample groups). Results of indicator species analysis (line 237) and DeSeq2 differential abundance analysis (line 238) have now been presented alongside these results in in line with the Reviewer’s comments. 

Discriminant analysis has many assumptions and restrictions, including equal sample sizes and homogeneity of variance/covariance that do not apply here.

4. The age should be one of important factors influencing oral microbial community. Given the wide range of years of children age (0.08-18.5 yeas), the authors should consider to divide the samples into different ago groups and compare the difference of bacterial composition and abundance.

We agree with reviewer 1, age is an important factor influencing the microbial community in this study. Several analyses were carried out to investigate the effect of age on the microbial community including controlling for age in the investigation of all clinical characteristics. The cohort of children below the age of 5 were investigated, only bacterial biomass was found to be significantly associated with age in this group. Further groups could not be analysed meaningfully due to the uneven distribution of the age data across the study.

5. Table 1 lacks statistical comparison of demographic information between cases and controls.

Statistical comparisons in the form of p-values have been added to Table 1 .

6. Figure 1 lacks the basic unit of Biomass. The percentage proportion of abundance should be indicated in the y-axis.

The units on Figure 1 have been updated to express the units of bacterial biomass.

7. Figure 2 - is it possible that the authors try other PCoA method to cluster samples, for example, weighted or unweighted UniFrac methods?

We thank reviewer one for suggesting alternative methods of clustering samples based on figure 2. Unweighted UniFrac relies on presences /absence data and does not take into account community composition which can be biased toward rare OTUs. On the other hand, weighted UniFrac uses phylogenetic information to infer genetic relationships based on evolutionary assumptions and this may lead to the introduction of errors. 

We have created alternative plots using these measures, but our conclusion is that Bray-Curtis dissimilarity is very commonly used for good reasons, and it remains the most easily understood measure for this plot.

8. The authors mentioned that "Specific care was made not to contaminate samples with any other part of the mouth". Was there more detailed information?

The protocols for performing oropharyngeal swabs have been updated with edits to both the methods and supplementary methods (page 7, line 135). 

Reviewer #2: 

The article is very relevant and evaluated viral respiratory infections and the oropharyngeal bacterial microbiota in acutely wheezing children. Although they concluded that the microbiota is not associated with viral infection and wheeze in children the article discuss important points that deserve to be published. 

We note with gratitude that the Reviewer considers this to be an interesting research study that would be improved by additions to the bioinformatic and biostatistical analyses. Our responses to specific points are:

1. It not clear if the children with bronchiolitis are infected with RSV or other virus. This must be explored in the results and in the conclusion. Other important point is to performed and analysis of family and order of the microbiota comparing children with bronchiolitis and the controls.

We have clarified the viral results associated with bronchiolitis diagnosis at the time of sampling in the results (line 274) and with reference to figure 1 in the manuscript.

2. Another point that should be explored in the manuscript is breastfeeding and Caesarean delivery, that might be influencing the microbiota.

We now state on page paragraph that “We did not find any differences in the microbiota that were attributable to recorded breastfeeding. No data was available to explore the effect of delivery method in this study.” Page 13, line 251.

Overall, we feel that the paper has been greatly improved by the Reviewers’ suggestions, and we trust these changes meet with your approval. 

Regards, 

Dr Leah Cuthbertson

---

## [Decision Letter · Decision Letter 1]

3 Oct 2019

Viral respiratory infections and the oropharyngeal bacterial microbiota in acutely wheezing children

PONE-D-19-20152R1

Dear Dr. Cuthbertson,

We are pleased to inform you that your manuscript has been judged scientifically suitable for publication and will be formally accepted for publication once it complies with all outstanding technical requirements.

With kind regards,

Dong-Yan Jin

Academic Editor

PLOS ONE

Additional Editor Comments (optional):

Reviewers' comments:

Reviewer's Responses to Questions

**Comments to the Author**

1. If the authors have adequately addressed your comments raised in a previous round of review and you feel that this manuscript is now acceptable for publication, you may indicate that here to bypass the “Comments to the Author” section, enter your conflict of interest statement in the “Confidential to Editor” section, and submit your "Accept" recommendation.

Reviewer #1: All comments have been addressed

Reviewer #2: (No Response)

2. Is the manuscript technically sound, and do the data support the conclusions?

Reviewer #1: Yes

Reviewer #2: Yes

3. Has the statistical analysis been performed appropriately and rigorously? 

Reviewer #1: Yes

Reviewer #2: Yes

4. Have the authors made all data underlying the findings in their manuscript fully available?

Reviewer #1: Yes

Reviewer #2: Yes

5. Is the manuscript presented in an intelligible fashion and written in standard English?

Reviewer #1: Yes

Reviewer #2: Yes

6. Review Comments to the Author

Reviewer #1: (No Response)

Reviewer #2: The authors improved the manuscript accepting the suggestions of the Reviewers. All my concerns had been addressed.

7. PLOS authors have the option to publish the peer review history of their article (what does this mean?). If published, this will include your full peer review and any attached files.

Reviewer #1: No

Reviewer #2: No

---

## [Editor Report · Acceptance letter]

7 Oct 2019

PONE-D-19-20152R1 

Viral respiratory infections and the oropharyngeal bacterial microbiota in acutely wheezing children 

Dear Dr. Cuthbertson:

I am pleased to inform you that your manuscript has been deemed suitable for publication in PLOS ONE. Congratulations! Your manuscript is now with our production department. 

With kind regards,

on behalf of

Prof. Dong-Yan Jin 

Academic Editor

PLOS ONE